# The fate of hippocampal synapses depends on the sequence of plasticity-inducing events

J Simon Wiegert[1,2], Mauro Pulin[1,2], Christine Elizabeth Gee[1], Thomas G Oertner[1]*

[1]Institute for Synaptic Physiology, Center for Molecular Neurobiology Hamburg, University Medical Center Hamburg-Eppendorf, Hamburg, Germany; [2]Research Group Synaptic Wiring and Information Processing, Center for Molecular Neurobiology Hamburg, University Medical Center Hamburg-Eppendorf, Hamburg, Germany

**Abstract** Synapses change their strength in response to specific activity patterns. This functional plasticity is assumed to be the brain's primary mechanism for information storage. We used optogenetic stimulation of rat hippocampal slice cultures to induce long-term potentiation (LTP), long-term depression (LTD), or both forms of plasticity in sequence. Two-photon imaging of spine calcium signals allowed us to identify stimulated synapses and to follow their fate for the next 7 days. We found that plasticity-inducing protocols affected the synapse's chance for survival: LTP increased synaptic stability, LTD destabilized synapses, and the effect of the last stimulation protocol was dominant over earlier stimulations. Interestingly, most potentiated synapses were resistant to depression-inducing protocols delivered 24 hr later. Our findings suggest that activity-dependent changes in the transmission strength of individual synapses are transient, but have long-lasting consequences for synaptic lifetime.

DOI: https://doi.org/10.7554/eLife.39151.001

*For correspondence:
thomas.oertner@zmnh.uni-hamburg.de

**Competing interests:** The authors declare that no competing interests exist.

## Introduction

Graded changes in synaptic strength, driven by specific activity patterns, are a candidate mechanism for information storage in the brain (*Chaudhuri and Fiete, 2016*). When entire pathways are potentiated by high frequency stimulation, the increase in synaptic coupling can indeed be recorded for several days (*Bliss and Lomo, 1973*). Increases in the size of synapses, the number of postsynaptic transmitter receptors and release of transmitter, have been shown to underlie increases in synaptic strength. A prevailing theory is that graded changes in synaptic strength persist as a memory trace of former activity. At the level of individual synapses, however, dramatic fluctuations in spine volume over time scales of hours to days cast doubt on whether information can be stored for long periods in the analog strength of synapses (*Holtmaat and Caroni, 2016*; *Berry and Nedivi, 2017*). An alternative hypothesis is that over longer time periods, information is stored not in the strength but in the number of connections, which, at the level of individual synapses, would manifest as a change in synaptic lifetime. Supporting evidence comes from the findings that long term depression (LTD) decreases synaptic lifetime (*Nägerl et al., 2004*; *Bastrikova et al., 2008*; *Wiegert and Oertner, 2013*) and that spine structure becomes stabilized and growth persists up to 3 days after induction of long term potentiation (LTP) (*De Roo et al., 2008*; *Hill and Zito, 2013*).

An important consideration is that new information, manifest as changing patterns of activity, constantly arrives at synapses. For example, LTP can be reversed by low-frequency stimulation (LFS), but such depotentiation may only occur 1–2 hr after LTP induction (*Fujii et al., 1991*; *O'Dell and Kandel, 1994*; *Abraham and Huggett, 1997*; *Zhou et al., 2004*). How a once potentiated synapse

responds to LFS one day later is therefore difficult to predict. Our goals were to monitor the fate of individual spine synapses after induction of LTP and to explore how sequential plasticity-inducing events affect synaptic lifetime. Using organotypic hippocampal slice cultures and optical stimulation of channelrhodopsin-expressing CA3 pyramidal neurons, we found that Schaffer collateral synapses were potentiated by 5 Hz stimulation if complex spike bursts were induced in the postsynaptic CA1 neuron (*Thomas et al., 1998*). We based our assessment of synaptic strength changes on the amplitude and probability of spine calcium transients (EPSCaTs). During successful synaptic transmission, $Ca^{2+}$ ions enter the spine through voltage-gated calcium cannels and NMDA receptors which both have a steep dependence on membrane depolarization. Thus, EPSCaTs depend on AMPA receptor activity (*Holbro et al., 2010*) and can be used to detect changes in synaptic strength (*Emptage et al., 2003*). Compared to glutamate uncaging experiments, which only report changes in postsynaptic strength (potency), optogenetic interrogation is also sensitive to presynaptic changes (release probability), providing a more complete picture of synaptic transmission. We then followed the fate of stimulated spine synapses and their neighbors over 7 days.

As suggested by previous studies, LTD and LTP differentially affected synaptic lifetime. However, sequentially inducing LTD and LTP did not return spines to their basal state, but resulted in reduced elimination rates similar to synapses which only underwent LTP. Once LTP was induced, it became almost impossible to induce subsequent LTD. In the few experiments were LTD could be induced 24 hr after LTP, synaptic lifetime was similar to that of spines that only underwent LTD. Thus, multiple weight adjustments are not summed in a linear fashion, but the most recent plasticity event determines the lifetime of a Schaffer collateral synapse.

## Results

### Optical theta frequency stimulation induced LTP at Schaffer collateral synapses

CA3 neurons expressing the light-sensitive channel ChR2(E123T/T159C) (*Berndt et al., 2011*) together with the presynaptic vesicle marker synaptophysin-tdimer2 were stimulated with short pulses of blue light (2 ms long, 40 ms interval, λ = 470 nm). Paired pulses were used to reduce the number of trials necessary to detect responding spines and to be consistent with our previous study (*Wiegert and Oertner, 2013*). On CA1 pyramidal cells expressing GCaMP6s and mCerulean, active spines were identified by imaging stimulation-induced excitatory postsynaptic calcium transients (EPSCaTs). After an active spine was identified we switched to line scanning mode, defining a scan curve that intersected the responding spine and a small number of neighboring spines at high speed (500 Hz, *Figure 1A*). Calcium transients were restricted to the responding spine and were not detected in the dendrite. To provide an additional read-out of synaptic strength on the population level, a neighboring CA1 cell ('reporter neuron') was patch-clamped to record excitatory postsynaptic synaptic currents (EPSCs) before, during, and after plasticity induction. Light stimulation evoked EPSCs with a magnitude of 1330 ± 220 pA, consistent with our previous study (*Wiegert and Oertner, 2013*). To induce LTP, we stimulated CA3 pyramidal cells with 150 light pulses at 5 Hz, a theta-frequency stimulation (TFS) paradigm, which potentiates CA3-CA1 but not CA3-CA3 synapses in an NMDAR-dependent fashion (*Moody et al., 1998*; *Thomas et al., 1998*). TFS-induced LTP requires transient (30 s) stimulation of enough CA3 cells to drive postsynaptic CA1 cells to fire complex spike bursts (CSBs, *Figure 1—figure supplement 1*) (*Thomas et al., 1998*). To facilitate LTP induction, ACSF with reduced divalent ion concentration (2 mM $CaCl_2$, 1 mM $MgCl_2$) was used to increase excitability. We adjusted the stimulation light intensity to recruit more and more CA3 neurons until the synaptic drive was just below the action potential threshold in the CA1 reporter neuron. During optogenetic theta-frequency stimulation (oTFS), the reporter neuron responses changed from mostly subthreshold EPSPs with occasional single action potentials to CSBs (*Figure 1C*, *Figure 1—figure supplement 1*) (*Losonczy and Magee, 2006*). CSBs in the reporter neuron were time-locked with large calcium transients in the stretch of dendrite adjacent to the postsynaptic spine (*Figure 1B*, middle column), suggesting that synchronized CSBs were occurring in neighboring neurons. EPSCaTs were strongly potentiated 30 min after oTFS, generating calcium transients that frequently spread into the dendrite (*Figure 1B*, *Figure 2B*). Likewise, the amplitude of EPSCs in the reporter neuron increased after oTFS, indicating successful induction of LTP (*Figure 1B*, *Figure 2A*). Both

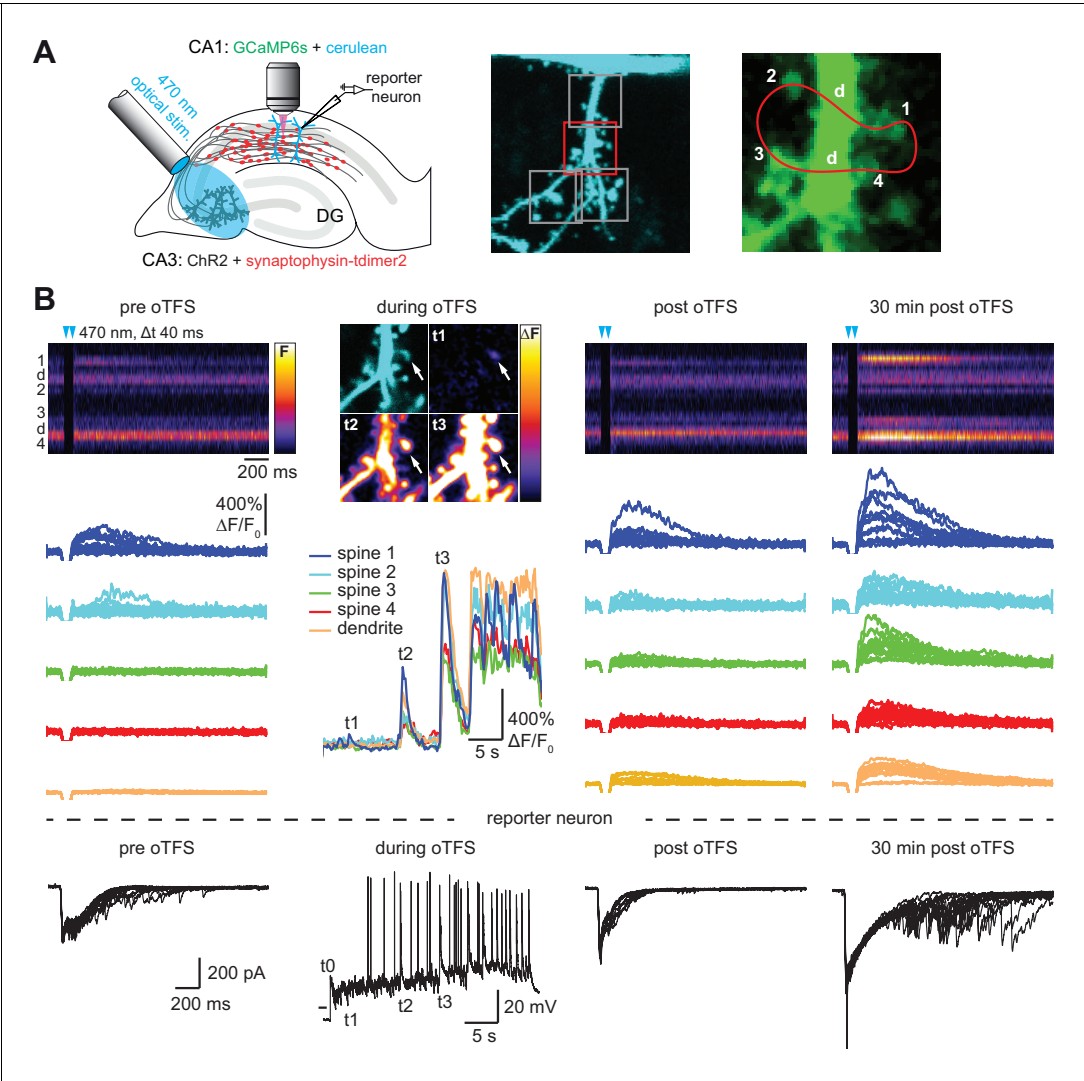

**Figure 1.** with two supplements: Channelrhodopsin-driven theta-frequency stimulation induces LTP. (**A**) Left: A fiber-coupled LED (λ = 470 nm) was used to locally stimulate ChR2-expressing CA3 neurons. Spines on GCaMP6s/mCerulean-expressing CA1 pyramidal cells were imaged with two-photon microscopy. For parallel electrical recordings, a second CA1 neuron was patch-clamped (reporter neuron). Middle: oblique dendrite branching off the apical trunk filled with mCerulean. Detection of active spines was done with GCaMP6s during presynaptic optogenetic stimulation. Stimulation-induced fluorescence changes (ΔF) of GCaMP6s were analyzed in fast frame scans (squares) of oblique dendrites until a responsive spine was detected (red square). Right: Magnified view of GCaMP6s fluorescence in the dendritic section harboring an activated spine. The laser was scanned in a user-defined trajectory across multiple spines and the parental dendrite during Ca2+ imaging (red curve). (**B**) Fluorescence signal across time from arbitrary line scan on dendrite shown in A during ChR2-stimulation before ('pre oTFS'), immediately ('post oTFS') and 30 min ('30 min post oTFS') after optical theta-frequency stimulation (oTFS). Temporally matched traces from multiple trials and electrophysiological recording from a reporter neuron are shown below. During oTFS the Ca2+ response was recorded in frame scan mode ('during oTFS'). The GCaMP6s-signal (ΔF) is shown for three selected time points during oTFS. GCaMP6s-traces from the same spines and dendrite imaged in line scans are shown below together with the corresponding electrophysiological recording in voltage clamp mode from the reporter neuron.

DOI: https://doi.org/10.7554/eLife.39151.002

The following figure supplements are available for figure 1:

**Figure supplement 1.** Examples of optogenetic TFS experiments.
DOI: https://doi.org/10.7554/eLife.39151.003
**Figure supplement 2.** Analysis of imaging and electrophysiology data.
DOI: https://doi.org/10.7554/eLife.39151.004

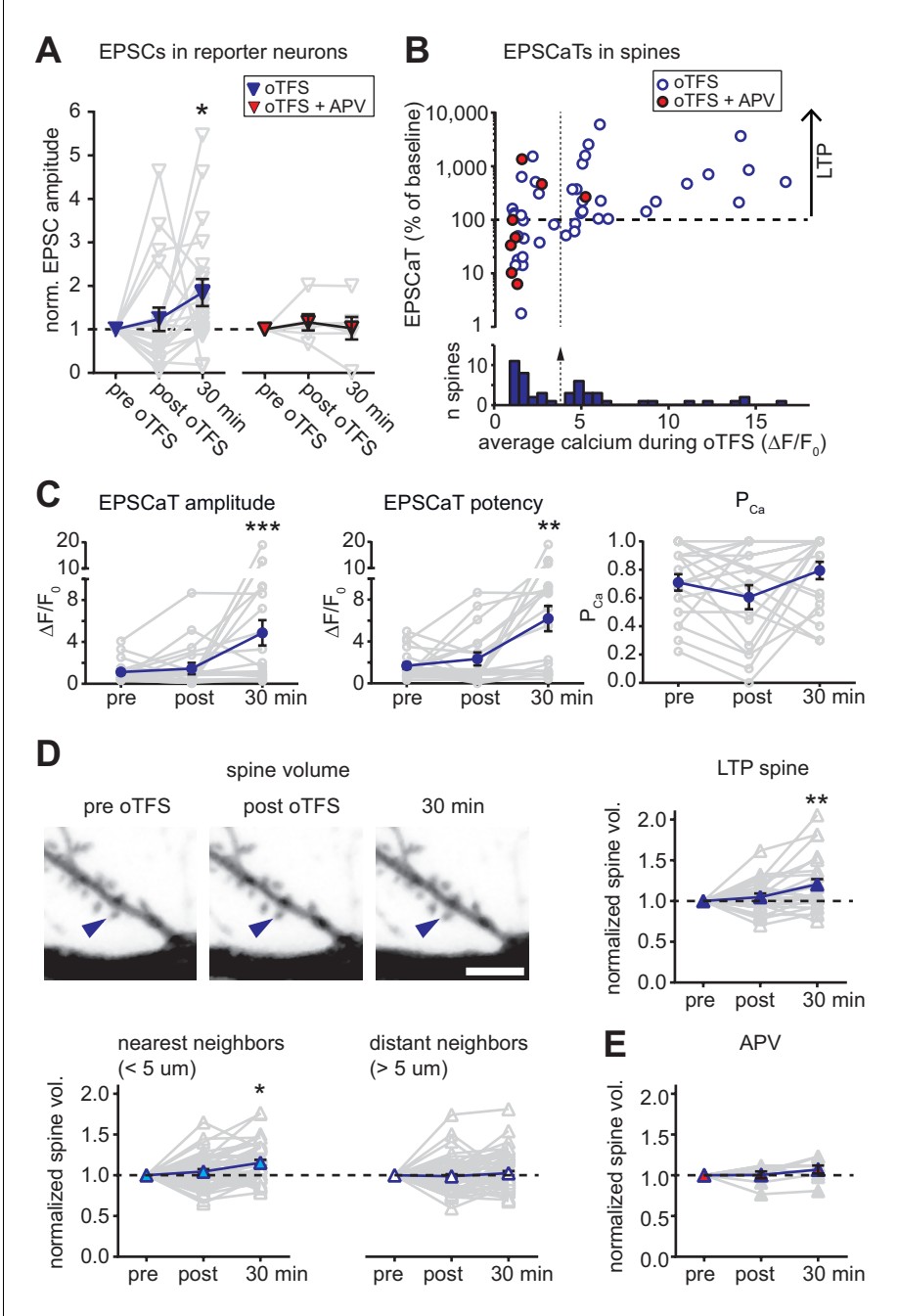

**Figure 2.** with two supplements: Characterization of oTFS-induced LTP. (**A**) Changes in excitatory postsynaptic current (EPSC) amplitude in reporter neurons immediately after and 30 min after oTFS in the absence (left) or presence (right) of the NMDA receptor antagonist APV during oTFS. EPSCs were significantly increased after 30 min (p=0.012, n = 20 slice cultures). The increase was blocked by APV (p=0.69, n = 6 slice cultures). (**B**) Relative change of average excitatory $Ca^{2+}$ transients (EPCaTs) in individual spines 30 min after the oTFS protocol plotted against the average spine $Ca^{2+}$ during oTFS. In experiments indicated by filled red circles, APV was present during oTFS. (**C**) EPSCaT amplitude (p=0.0008, n = 20 slice cultures) and EPSCaT potency (successes only, p=0.0025) but not EPSCaT probability ($P_{Ca}$, p>0.05) were increased 30 min after oTFS in experiments where complex spike bursts (CSBs) were induced during oTFS. (**D**) Maximum intensity projections of mCerulean fluorescence in dendritic segment harboring a responding spine that was successfully potentiated (blue arrowhead). Volume of oTFS spines (p=0.002, n = 26 spines) and nearest (p=0.0001, n = 45 spines) but not distant neighbors (p=0.83, n = 58 spines) was increased 30 min after oTFS in experiments where CSBs were induced during oTFS. (**E**) Spine volume was not increased when NMDA receptors were blocked with APV during oTFS (p>0.05, n = 7 spines).

*Figure 2 continued on next page*

*Figure 2 continued*

DOI: https://doi.org/10.7554/eLife.39151.005

The following source data and figure supplements are available for figure 2:

**Source data 1.** Theta-frequency stimulation experiments.

DOI: https://doi.org/10.7554/eLife.39151.008

**Figure supplement 1.** Analysis of oTFS experiments where no dendritic calcium spikes were observed during the induction protocol.

DOI: https://doi.org/10.7554/eLife.39151.006

**Figure supplement 2.** Analysis of control experiments where no oTFS was applied to responding spines.

DOI: https://doi.org/10.7554/eLife.39151.007

CSBs and LTP induction were blocked in the presence of the NMDA receptor antagonist APV (*Figure 2A and B*). Thus, oTFS induced plasticity via NMDAR activation as previously demonstrated (*Thomas et al., 1998*). In some experiments, CSBs and large dendritic calcium transients did not occur during oTFS, likely due to insufficient numbers of virus-transfected CA3 pyramidal neurons. When no large dendritic calcium transients were triggered during oTFS, spine calcium signals were not consistently potentiated 30 min after oTFS (*Figure 2B*, *Figure 2—figure supplement 1A and B*). To estimate changes in EPSCaT and EPSC amplitude, 10–20 successive traces before and after stimulation were analyzed and averaged (*Figure 1—figure supplement 2*). EPSCs, integrating the activity of many synapses, showed considerably lower trial-to-trial variability (no failures) compared to EPSCaTs.

Taking into consideration only those experiments in which CSBs and dendritic calcium transients were evoked during oTFS, we observed that neither the amplitude nor the potency (amplitude of successes) of EPSCaTs changed immediately after oTFS. Thirty minutes later, however, both were significantly increased (*Figure 2C*), whereas spines that did not experience oTFS showed no change in EPSCaTs over time (*Figure 2—figure supplement 2*). The slowly developing potentiation was also reflected in the EPSCs recorded in the reporter neuron (*Figure 2A*), consistent with previous reports (*Moody et al., 1998*; *Thomas et al., 1998*). LTP had no significant effect on the probability of EPSCaT occurrence ($P_{Ca}$, *Figure 2C*), suggesting that the potentiation was mainly due to postsynaptic changes. Interestingly, while EPSCaT potency was not affected in experiments where no CSBs were elicited during oTFS, $P_{Ca}$ was significantly reduced (*Figure 2—figure supplement 1C*). Thus, in experiments where the synaptic drive was not strong enough to trigger postsynaptic spikes, presynaptic activity in the theta frequency range appeared to elicit a weak form of presynaptic depression. No such reduction in $P_{Ca}$ was seen when no oTFS was applied (*Figure 2—figure supplement 2*).

Optogenetic TFS-induced synaptic potentiation was accompanied by slow changes in spine structure (mCerulean, *Figure 2D*). The head volume of spines that experienced CSBs was unchanged immediately after oTFS, but increased by 21 ± 6% during the next 30 min. The nearest neighboring spines also showed a small but significant increase in volume (15 ± 3%), whereas no consistent change was detected at more distant spines (2 ± 3%). When no oTFS was applied, the volume of responding spines (5 ± 4%) and neighbors (−4 ± 3%) remained stable (*Figure 2—figure supplement 2*). In oTFS experiments that failed to elicit CSBs or when NMDA receptors were blocked during oTFS, stimulated spines did not exhibit significant volume changes (*Figure 2E*; *Figure 2—figure supplement 1D*), suggesting similar requirements for the successful induction of functional and structural plasticity. As our functional assessment was limited to the few spines that were synaptically connected to ChR2-expressing CA3 neurons, we could not test whether neighboring enlarged spines were also functionally potentiated.

## Synaptic properties 24 hr after LTP

We next asked whether synaptic potentiation was maintained during the 24 hr following oTFS. Consistent with our first data set, spine head volume and EPSCaT potency were significantly increased 30 min after oTFS (*Figure 3A and B*). Twenty-four hours after LTP induction, however, both measures had returned to baseline. We detected no significant change in EPSCaT probability either 30 min or 24 hr after oTFS (*Figure 3C*). Thus, beyond the acute effects on the day of potentiation, we did not observe permanent changes in synaptic strength after oTFS-induced LTP.

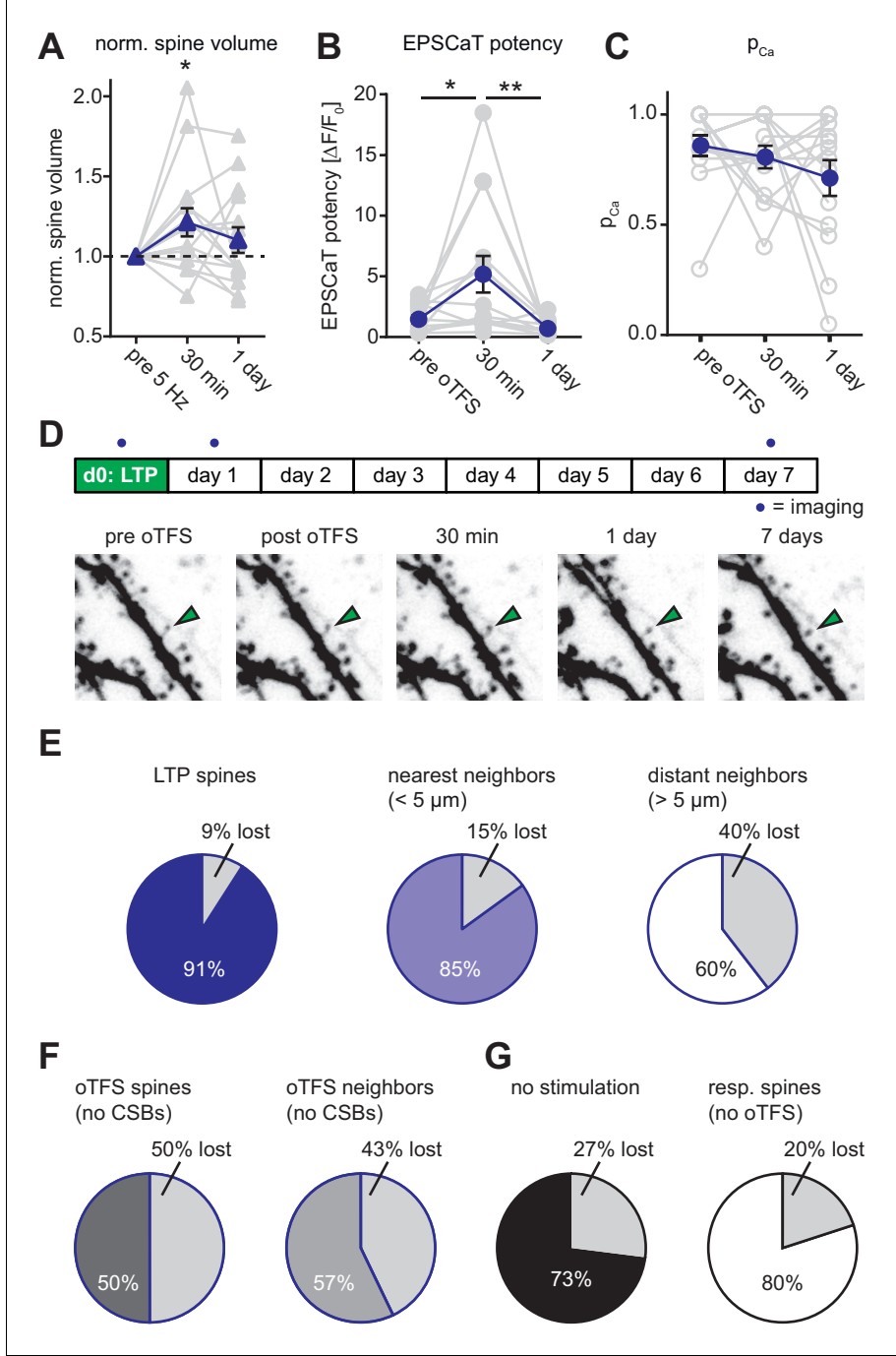

**Figure 3.** Long-term outcome of oTFS-induced LTP. (**A**) Analysis of volume changes of oTFS spines 30 min and 24 hr after oTFS. The volume increase 30 min after oTFS (p=0.03, n = 15 slice cultures) was not maintained 24 hr later (p=0.42). (**B**) Analysis of EPSCaT potency before, 30 min and 24 hr after oTFS. The increased potency 30 min after oTFS (p=0.015, n = 14 slice cultures) has significantly decreased again 24 hr later (p=0.005) and was similar to the condition before oTFS (p=0.55). (**C**) EPSCaT probability ($P_{Ca}$) did not change 30 min and 24 hr after oTFS (p=0.32, n = 14 slice cultures). For details on the statistical tests, please refer to the Materials and Methods section. (**D**) Long-term survival analysis after LTP. Spines were imaged at d0, d1 and d7. Maximum intensity projections of mCerulean fluorescence in dendritic segment harboring a responding spine that was successfully potentiated (green arrowhead). (**E**) Spine survival 7 days after successful LTP induction on day 0. Surviving fractions are shown for responding spines, nearest and distant neighbors. (**F**) Spine survival 7 days after oTFS in experiments where no complex spike bursts were induced. Directly stimulated spines and their neighbors were analyzed separately. (**G**)

*Figure 3 continued on next page*

*Figure 3 continued*

Spine survival over 7 days under baseline conditions without any optical stimulation (black) and in spines responsive to optical test pulses (resp. spines, white) which were not exposed to plasticity-inducing protocols.
DOI: https://doi.org/10.7554/eLife.39151.009

The following source data is available for figure 3:

**Source data 1.** Theta-frequency stimulation: Spine volume changes.
DOI: https://doi.org/10.7554/eLife.39151.010

## Effect of long-term potentiation on synaptic lifetime

Next, we determined whether oTFS-induced LTP affected synaptic lifetime. Previous work showed that potentiated spines are not characterized by permanently enlarged heads, but are less likely to be eliminated during the next 3 days (*De Roo et al., 2008*). We therefore assessed the stability of potentiated spines and their neighbors during the following week. Under control conditions without external stimulation, 27% of all spines disappeared between days 1 and 7. This turnover rate is in agreement with previous measurements in hippocampal slice cultures (*Wiegert and Oertner, 2013*) and mouse hippocampus in vivo (*Attardo et al., 2015*). LTP induced by oTFS appeared to increase synaptic lifetime. In a dataset of 14 spines, 11 spines experienced CSBs and were potentiated. During the following 7 days, only one of these 11 potentiated spines disappeared (*Figure 3D and E*). The stability of spines next to the potentiated spine was also affected, mirroring the transient head volume increase on day 0 (*Figures 2D* and *3A*). Compared to controls, nearest-neighbor spines disappeared less often between days 1 and 7 whereas more distant spines (>5 µm) were eliminated more often (*Figure 3E*). These findings are consistent with the concept of biochemical signaling molecules activated inside stimulated spines during oTFS and diffusing into neighboring, non-stimulated spines (*Nishiyama and Yasuda, 2015*), affecting acutely their size and on longer timer scales, their survival. As a control, we also analyzed oTFS experiments in which no CSBs were elicited in CA1 neurons. In these experiments, stimulated spines as well as their neighbors had reduced survival rates (*Figure 3F*). This destabilizing effect was contingent on 5 Hz presynaptic activation, as spines that were not stimulated at all or only stimulated by test pulses (responsive spines) had higher survival rates (*Figure 3G*).

## Effects of sequential plasticity-inducing protocols on synaptic lifetime

As we established previously (*Wiegert and Oertner, 2013*), optogenetic low frequency stimulation (oLFS, 900 APs at 1 Hz) induced long-term depression (LTD) at Schaffer collateral synapses, frequently abolishing EPSCaTs in the stimulated spine altogether (*Figure 4A*). In agreement with our previous results, 45% of spines that received oLFS disappeared between days 1 and 7. We speculated that if we induced LTP 24 hr after LTD, the doomed spines could perhaps be stabilized (*Figure 4B*). LTD on day 0 was considered successful when the average spine $Ca^{2+}$ response dropped to less than 90% of the baseline response 30 min after oLFS, which was the case in 70% (28/40) of the experiments (*Figure 4—figure supplement 1*). Twenty-four hours later, we recorded a new baseline, since EPSCaT amplitudes frequently changed from one day to the other. We then applied oTFS to the spines that were depressed on the previous day. LTP induction was considered successful when the average spine $Ca^{2+}$ response increased to more than 110% of the day one baseline response after oTFS, which was the case in 64% (18/28) of all experiments on day 1 (*Figure 4C*). We also considered spines that did not experience CSBs to assess whether the oTFS protocol itself would affect synapse lifetime independently of successful LTP induction. Thus, we compared two groups: synapses that underwent LTD followed by LTP (45% of all tested synapses) and synapses that also experienced LTD and oTFS, but did not show any potentiation in response to oTFS (25% of all tested synapses). Synapses that did not display LTD after oLFS on day 0 (30% of all tested synapses) were not considered further. When LTP was induced after LTD, only 12% of spines disappeared between days 1 and 7, indicating stabilization of doomed synapses. Of the spines that received oTFS after LTD but did not get potentiated, 43% disappeared between days 1 and 7, similar to the 45% disappearance rate seen after oLFS only. These results confirmed that successful LTP induction was necessary to rescue synapses from elimination. Without LTP, the oTFS stimulation protocol by itself had no measurable effect on the survival of previously depressed synapses.

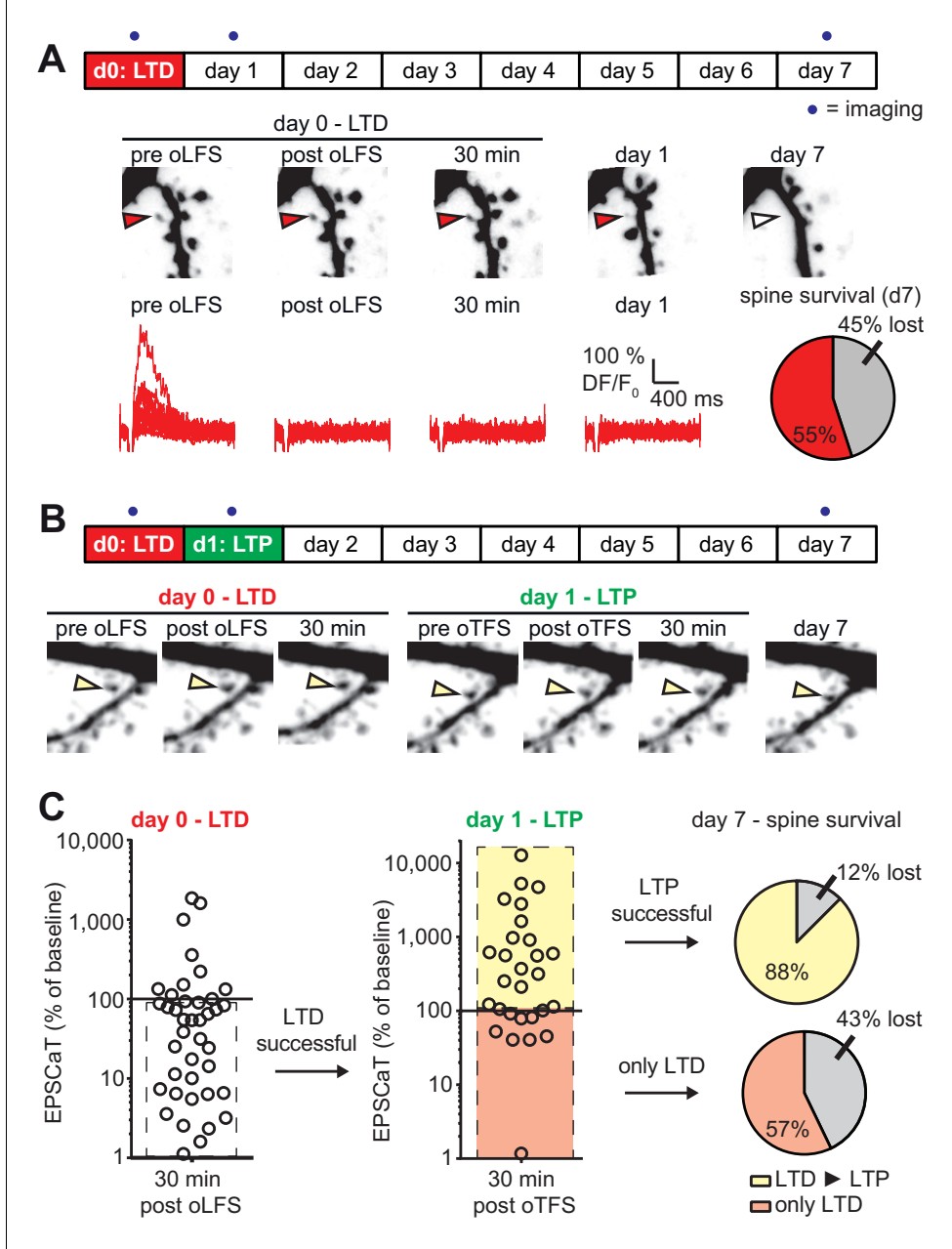

**Figure 4.** with one supplement: LTD-induced spine elimination is reversed by LTP or sustained synaptic transmission. (**A**) Long-term survival analysis after LTD. Spines were imaged at d0, d1 and d7. Below: Maximum intensity projections of mCerulean fluorescence in dendritic segment harboring a responding spine that was successfully depressed (red arrowhead). Open arrowhead on day seven indicates position of eliminated spine. Corresponding EPSCaT traces from indicated time points are shown in red. Pie chart shows quantification of spine survival after 7 days. (**B**) LTP 24 hr after LTD. Below: Dendritic segment harboring a responding spine that was successfully depressed on day 0 and potentiated on day 1 (yellow arrowhead). (**C**) Assessment of synaptic weight changes induced by oLFS on day 0 and oTFS on day 1. Dashed box in left graph indicates all experiments where LTD was successfully induced on day 0. Only these spines were considered in the LTP experiment on day 1 (middle). Yellow shaded box indicates all experiments where LTP was successfully induced on day 1 (after LTD on day 0; LTD ▶ LTP). Red shaded box indicates experiments where oTFS did not lead to LTP (only LTD). Pie charts show quantification of spine survival after 7 days for these two conditions.

DOI: https://doi.org/10.7554/eLife.39151.011

The following source data and figure supplement are available for figure 4:

**Source data 1.** Low-frequency stimulation followed by theta-freuquency stimulation.

*Figure 4 continued on next page*

*Figure 4 continued*

DOI: https://doi.org/10.7554/eLife.39151.013
**Figure supplement 1.** LTD followed by LTP.
DOI: https://doi.org/10.7554/eLife.39151.012

We next tested whether the LTP-induced stabilization of spines would persist even if LTD was subsequently induced (*Figure 5A*). LTP on day 0 was induced in 72% (18/25) of spines after oTFS (*Figure 5B*, *Figure 5—figure supplement 1*), similar to the set of oTFS experiments 1 day after oLFS (64%, p=0.85) and the first set of oTFS experiments (*Figure 2B*, 62%, 26/42, p=0.41). Again, we considered only spines where LTP was successfully induced. When oLFS was applied 24 hr later, we observed that LTD was induced in only 33% (6/18) of previously potentiated synapses on the next day (*Figure 5B*), a much lower success rate than the 70% when oLFS was applied with no prior plasticity. Since we were concerned that the 1 Hz induction protocol could have become supra-threshold 24 hr after LTP, we counted the number of spikes generated in the reporter neuron during oLFS. The median number of APs in reporter neurons during the 900 pulses of the oLFS protocol was 1.5 in naive cultures (n = 32) and 5.0 one day after LTP (n = 13, p=0.4, Mann-Whitney), corresponding to a postsynaptic spike probability below 1% in both cases. Thus, strong postsynaptic spiking during the oLFS protocol is not a likely explanation for the difficulty to depress previously potentiated synapses. We also considered the possibility that some synapses were already in a depressed state and could therefore not be depressed further. However, the initial EPSCaT amplitude (before oLFS) was not a predictor of successful LTD induction (*Figure 5—figure supplement 1C*). These results point to a synapse-specific memory of past potentiation events that cannot be detected as increased spine volume, increased release probability or increased EPSCaT potency (see *Figure 3*).

In the few experiments where LTD was successfully induced 24 hr after LTP, 50% of spines disappeared by day 7 (*Figure 5B*). In the more typical case where oLFS failed to induce LTD, only 8% of spines disappeared by day 7. One explanation for the different survival rates could be that the absolute strength of the synapse before the oLFS protocol determined whether it survived, irrespective of the sign of plasticity on day 1 (i.e. the synapse has a memory of its strength and not of its plastic change). However, the strength of the synapse on day 0 or on day one did not predict its survival (*Figure 5—figure supplement 1D and E*), leaving successful induction of depression as the only risk factor we could identify. In summary, the stabilizing effect of LTP on spines can be overwritten by subsequent LTD (*Figure 5C*), but this sequence of plasticity events is not very likely to happen.

## Discussion

In vitro studies of synaptic plasticity are most relevant if stimulation protocols resemble in vivo activity patterns. Theta burst stimulation (TBS, 100 Hz bursts repeated at 5 Hz) is a commonly used experimental protocol to induce LTP in vitro (*Abraham and Huggett, 1997*), but individual CA3 pyramidal cells do not spike at 100 Hz in vivo (*Mizuseki and Buzsáki, 2013*). During exploratory behavior, CA3 pyramidal cells fire single action potentials which are synchronized across the population by the activity of local interneurons. Here we show that LTP and spine-specific stabilization can be induced at 5 Hz, the typical carrier frequency of rodent hippocampus, if a sufficient number of inputs are activated synchronously (*Moody et al., 1998*; *Thomas et al., 1998*). We consider theta-frequency stimulation (TFS) the physiological equivalent of spike-timing-dependent potentiation (STDP) protocols, replacing the artificial current injection into the postsynaptic neuron by highly synchronized excitatory synaptic input. Synchronized synaptic input can trigger dendritic calcium spikes, local regenerative events caused by the opening of voltage-dependent channels (NMDARs and VDCCs). These events can be electrophysiogically identified as complex spike bursts, consisting of several fast sodium spikes on top of a broader depolarization mediated by dendritic calcium currents (*Magee and Johnston, 1997*; *Golding et al., 2002*; *Losonczy and Magee, 2006*; *Grienberger et al., 2014*). In our experiments, the occurrence of dendritic calcium spikes during the induction protocol was highly predictive of successful LTP induction at individual synapses (*Figure 2B*). Recent studies in head-fixed mice running on a treadmill suggest that theta-frequency-modulated synaptic input to CA1 pyramidal cells triggers dendritic calcium spikes which are required

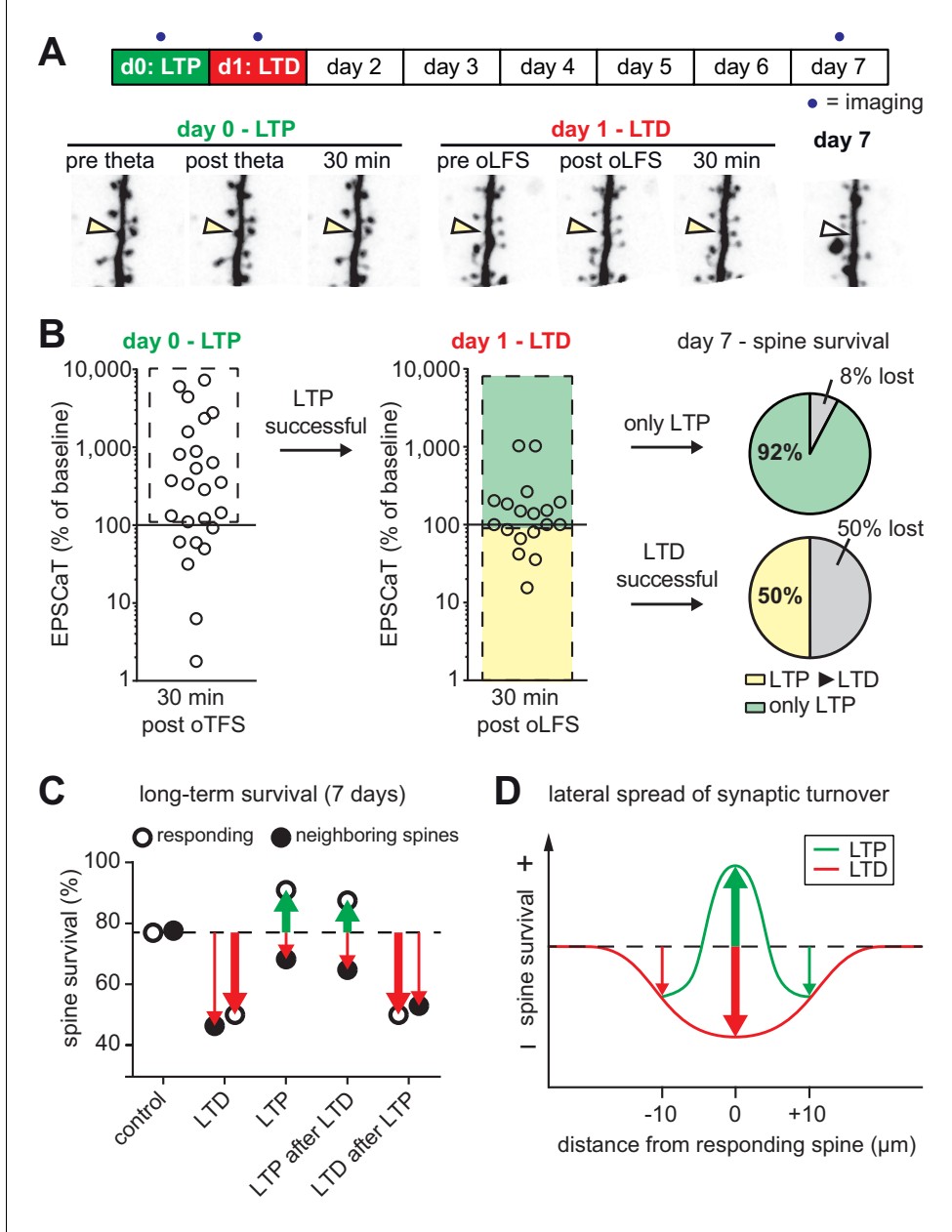

**Figure 5.** with one supplement: The most recent plasticity event fully accounts for synaptic tenacity. (**A**) Long-term survival analysis of experiments where LTD was induced 24 hr after LTP. Maximum intensity projections of mCerulean fluorescence in dendritic segment harboring a responding spine that was successfully potentiated on day 0 and depressed on day 1 (yellow arrowhead). (**B**) Assessment of synaptic weight changes induced by oTFS on day 0 and oLFS on day 1. Dashed box in left graph indicates all experiments where LTP was successfully induced on day 0. Only these spines were considered in the LTD experiment on day 1 (middle). Yellow shaded box indicates all experiments where LTD was successfully induced on day 1 (after LTP on day 0, LTP ▶ LTD). Note the low probability of depression after potentiation. Green shaded box encompasses experiments where oLFS did not lead to LTD or even led to LTP (only LTP). Pie charts show quantification of spine survival after 7 days for these two conditions. (**C**) Comparison of spine survival 7 days after various plasticity paradigms. Stimulated spines are shown as open circles; non-stimulated neighbors within 10 µm are shown as filled circles. Values for 'control' and 'LTD' are from *Wiegert and Oertner, 2013*. (**D**) LTP stabilizes the spine carrying the potentiated synapse, but reduces the average lifetime of more distant (>5 µm) spines on the same dendrite.
DOI: https://doi.org/10.7554/eLife.39151.014

The following source data and figure supplement are available for figure 5:

*Figure 5 continued on next page*

*Figure 5 continued*

**Source data 1.** Theta-frequency stimulation followed by low-freuquency stimulation.
DOI: https://doi.org/10.7554/eLife.39151.016
**Figure supplement 1.** LTP followed by LTD.
DOI: https://doi.org/10.7554/eLife.39151.015

for synaptic potentiation and place cell formation (*Bittner et al., 2015*; *Sheffield et al., 2017*). Thus, dendritic calcium spikes during complex spike bursts, evoked by synchronized input from entorhinal cortex and CA3 pyramidal cells, are part of the physiological mechanism for the selective potentiation of active Schaffer collateral synapses during behavior (*Hasselmo et al., 2002*).

Spine calcium imaging allowed us to detect synaptic plasticity at single synapses without electrodes. Newly inserted AMPA receptors lead to stronger depolarization of the spine head during the EPSP, more efficient unblocking of NMDA receptors and EPSCaT potentiation. It is important to note, however, that EPSCaT amplitudes are not linearly related to somatic EPSCs. The ratio between AMPA and NMDA receptors is not constant between spines, and peak calcium concentrations depend on spine head volume and spine neck resistance (*Grunditz et al., 2008*). High EPSCaT amplitudes can even lead to SK channel activation and dampening of the EPSP (*Bloodgood and Sabatini, 2007*). These confounds, which make EPSCaT amplitude comparisons between spines difficult, are less of a problem when the same spine is compared before and after plasticity induction to differentiate between LTP and LTD.

Counting the number of EPSCaTs in a set of stimulated trials can be used as a proxy for presynaptic release probability, as postsynaptic failures (successful glutamate release without postsynaptic calcium influx) are thought to be rare at Schaffer collateral synapses (*Nimchinsky et al., 2004*). In contrast to LTD, where the reduction in average EPSCaT amplitude was mainly due to decreased release probability (*Wiegert and Oertner, 2013*), oTFS-induced LTP strongly enhanced EPSCaT potency, but did not seem to affect release probability. This confirms that postsynaptic mechanisms such as AMPA receptor insertion account for this form of potentiation (*Shi et al., 1999*; *Lu et al., 2001*; *Matsuzaki et al., 2004*). Analyzing spine volume changes supported the notion of pre- vs postsynaptic plasticity mechanisms: While LTD induction did not affect spine volume (*Wiegert and Oertner, 2013*), LTP triggered significant growth of the postsynaptic compartment (*Figure 2D*). Going beyond the first hours after plasticity induction, we asked how these different forms of plasticity would influence the tenacity of synapses that actively contributed to postsynaptic spiking in comparison to inactive synapses on the same dendrite.

Twenty-four hours after induction of LTP, synapses were back to their baseline state with respect to the amplitude and probability of spine calcium transients as well as the volume of the spine head. Yet, a long-lasting, synapse-specific memory of the potentiation event was maintained, since these once-potentiated spines were more likely to persist during the following week compared to other spines on the same dendritic branch, or non-stimulated controls. Similarly, the effects of LTD may outlast the actual depression: CA1 spines did not show any lasting reduction in volume, but their life expectancy was significantly reduced after LTD (*Wiegert and Oertner, 2013*). A similar sequence of transient LTD followed by delayed spine elimination was found at the parallel fiber synapse on Purkinje cells in the cerebellum (*Aziz et al., 2014*). These findings support the theoretical concept that information could be robustly stored in the topology of the network rather than in the analog strength of individual synapses. The mechanism linking LTP to synaptic stabilization, and LTD to destabilization, is likely to involve several processes. Synaptic tenacity is known to be affected by trans-synaptic proteins such as Neuroligin-1 and SynCAM-1 (*Zeidan and Ziv, 2012*; *Körber and Stein, 2016*), PSD-95 (*De Roo et al., 2008*; *Cane et al., 2014*), ubiquitin protein ligase E3A (*Kim et al., 2016*), ensheathment of the synapse by astrocyte processes (*Bernardinelli et al., 2014*) and many other local factors. It may be a combination of local physical changes and distributed network effects, such as the recurrent reactivation of a specific circuit (*Wei and Koulakov, 2014*; *Novitskaya et al., 2016*), which makes once potentiated synapses robust against depression (*Figure 5B*) and pruning (*Figure 3E*). New tools for chronic activity modulation may allow dissecting use-dependent synapse stabilization in future experiments (*Lopez et al., 2016*; *Beck et al., 2018*). The link between LTP and long-term structural stability we show on the single-synapse level could explain why learning-induced spines in motor cortex are more stable than their pre-existing

neighbors and persist for months after training (*Xu et al., 2009*; *Yang et al., 2009*). LTP-induced tenacity might be a general principle to connect different time scales of cortical circuit plasticity.

Failure to evoke postsynaptic CSBs upon oTFS led to presynaptic depression in our experiments, which was followed by increased spine elimination (*Figure 3F*). This effect has been shown to be mediated by autocrine glutamate signaling at the presynaptic terminal and may not involve postsynaptic signaling (*Padamsey et al., 2017*). If, on the other hand, the postsynaptic neuron is driven to spike, retrograde signaling via NO (nitric oxide) leads to an increase in release probability, which explains why we did not see presynaptic depression in synapses that experienced CSBs (*Figure 2C*). Thus, the classical Hebbian rule of rewarding only synapses that causally contribute to postsynaptic AP firing also seems to apply to long-term stability. However, our results suggest that changes are not perfectly confined to the directly driven synapse: Optogenetic TFS not only affected strength, volume and long-term stability of the stimulated spines, but also increased volume and stability of its immediate neighbors (*Figure 2D*, *Figure 3E*). This is consistent with short-range diffusion of 'potentiating factors' such as activated RhoA and Cdc42 out of the directly stimulated spine (*Murakoshi et al., 2011*; *Yasuda, 2017*). In contrast, more distant spines on the same dendrite (5–10 μm) showed no increase in volume and a decrease in lifetime (*Figure 2D*, *Figure 3E*), confirming an earlier 3 day study (*De Roo et al., 2008*). Since we increased the optogenetic drive to CA3 during oTFS, we could not map the position of all spines that were active during plasticity induction. Therefore, we were not able to study the spatial extent of spine destabilization, for example by selecting a 'control' branch that received no input during oTFS. Nevertheless, our 7 day follow-up points to a center-surround function that stabilizes the immediate neighbors (<5 μm) of potentiated synapses, although they were most likely not active during the induction protocol (*Figure 5D*). As we have previously shown, LTD-induced destabilization has an even larger (>10 μm) lateral spread (*Wiegert and Oertner, 2013*). Apparently, the local environment is as important for the long-term survival of a synaptic connection as its own activity history. This could put a limit to the uniformity of synaptic inputs in dendritic sections, as it might be impossible to prune a synapse next to a strongly potentiated spine.

By inducing two rounds of plasticity, we demonstrated that synaptic pruning is not a random process, but determined by the last plasticity-inducing activity pattern. In the organotypic culture system, the latency between LTD induction and spine loss was several days. This period could be considerably shorter in vivo, given the highly rhythmic activity of the hippocampal circuit and in consequence, intense synaptic competition. Our approach allows imposing any kind of spike pattern to a select group of synapses over several days. It complements in vivo studies of structural plasticity, which provide information about spine turnover, but not about the activity patterns in pre- and postsynaptic neurons (*Attardo et al., 2015*). Once the conditions for synaptic maintenance are understood, the protracted process of circuit refinement by constant removal of irrelevant synapses could be simulated. Networks with self-organized connectivity might generate activity patterns that are different from the randomly connected networks underlying current large-scale simulations (*Markram et al., 2015*). Together with realistic simulations of synaptic network dynamics and long-term investigations of synapse remodeling in vivo, long-term analysis of the structure-function relationship of individual synapses may help understanding how the brain stores and retrieves memories.

# Materials and methods

## Key resources table

| Reagent type (species) or resource | Designation | Source or reference | Identifiers | Additional information |
|---|---|---|---|---|
| Strain, strain background (Rattus norvegicus, male) | Wistar | Charles River | Crl:WI | bred in the animal facility, UKE Hamburg |

*Continued on next page*

*Continued*

| Reagent type (species) or resource | Designation | Source or reference | Identifiers | Additional information |
|---|---|---|---|---|
| Strain, strain background (R. norvegicus, male) | Wistar | Janvier | RjHAN:WI | bred in the animal facility, UKE Hamburg |
| Genetic reagent (Clamydomonas reinhardtii) | ChR2(ET/TC) | doi: 10.1073/pnas.1017210108 | | channelr hodopsin |
| Genetic reagent (Aequorea victoria) | GCaMP6s | doi: 10.1038/nature12354 | | calcium indicator |
| Genetic reagent (A. victoria) | mCerulean | doi: 10.1038/nbt945 | | fluorescent protein |
| Transfected construct (R. norvegicus) | ChR2(ET/TC)−2A-synaptophysin-tdimer2 | doi: 10.1073/pnas.1315926110 | | transfection of CA3 neurons |
| Recombinant DNA reagent | rAAV2/7 | Vector Facility UKE Hamburg | | viral vector |
| Chemical compound, drug | APV | Tocris Bioscience | CAS Number 79055-68-8 | NMDA receptor blocker |
| Software, algorithm | ScanImage3.8 | DOI: 10.1186/1475-925X-2−13 | | modified for arbitrary line scans |

## Slice culture preparation and transfection

Hippocampal slice cultures from male Wistar rats were prepared at postnatal day 4–5 as described (*Gee et al., 2017*). Animal procedures were in accordance with the guidelines of local authorities and Directive 2010/63/EU. At DIV 3, we pressure-injected rAAV2/7 encoding ChR2(ET/TC)−2A-synaptophysin-tdimer2 into CA3. At DIV 18, single-cell electroporation was used to transfect CA1 pyramidal neurons in rAAV-infected slices with GCaMP6s and mCerulean (ratio 1:1) as described (*Wiegert et al., 2017*).

## Electrophysiology

Experiments were performed between DIV 21 and 25. Whole-cell recordings from CA1 pyramidal cells were made at 25°C with a Multiclamp 700B amplifier (Molecular Devices). Patch pipettes with a tip resistance of 3–4 MΩ were filled with (in mM) 135 K-gluconate, 4 $MgCl_2$, 4 $Na_2$-ATP, 0.4 Na-GTP, 10 $Na_2$-phosphocreatine, three ascorbate, and 10 HEPES (pH 7.2). LTD experiments were conducted in ACSF containing (in mM) 135 NaCl, 2.5 KCl, 4 $CaCl_2$, 4 $MgCl_2$, 10 Na-HEPES, 12.5 D-glucose, 1.25 $NaH_2PO_4$, 0.03 D-Serine (pH 7.4, sterile filtered). During LTP induction, ACSF with lower divalent ion concentration (2 $CaCl_2$, 1 $MgCl_2$) was used to increase excitability. Access resistance was monitored continuously and recordings with a drift of >20% were discarded.

## Two-Photon microscopy

The custom-built two-photon imaging setup was based on an Olympus BX51WI microscope equipped with a LUMPLFLN 60 × 1.0 NA objective, controlled by the open-source software package ScanImage (*Pologruto et al., 2003*) which was modified to allow user-defined arbitrary line scans at 500 Hz. Two Ti:Sapphire lasers (MaiTai DeepSee, Spectra-Physics) controlled by electro-optic modulators (350–80, Conoptics) were used to excite cerulean (810 nm) and GCaMP6s (980 nm). To activate ChR2(ET/TC)-expressing cells outside the field of view of the objective, we used a fiber-coupled LED (200 µm fiber, NA 0.37, Mightex Systems) to deliver light pulses to CA3. During the blue light pulses, sub-stage PMTs (H7422P-40SEL, Hamamatsu) were protected by a shutter (NS45B, Uniblitz).

## Measuring excitatory postsynaptic calcium transients (EPSCaTs)

Frame scans (10 × 10 μm) of oblique dendrites were acquired to detect spines responding to opto-genetic stimulation of CA3 neurons. Two brief (2 ms) light pulses with an inter-pulse interval of 40 ms were applied to increase release probability and thus the chance of detecting responding spines. In each trial, 14 frames (64 × 64 pixel) were acquired at 7.8 Hz. At least five trials were recorded from each dendritic segment. The relative change in GCaMP6s fluorescence ($\Delta F/F_0$) was calculated on-line. If the spine signal exceeded two times the standard deviation (SD) of its resting fluorescence, this spine was considered as 'potentially responding'. To measure $Ca^{2+}$ transients with better signal-to-noise ratio, line scans were acquired across potentially responding spine heads and their parent dendrites (500 Hz, 20 trials/spine). To measure the amplitude of $Ca^{2+}$ transients and to distinguish successful synaptic transmission events (EPSCaTs) from failures, we used a template-based fitting algorithm. The characteristic fluorescence time constant was extracted for every spine by fitting a double exponential function ($\tau_{rise}$, $\tau_{decay}$) to the average GCaMP6s signal. To estimate the $Ca^{2+}$ transient amplitude for every trial, we fit the spine-specific template to every response, amplitude being the only free parameter. Response amplitude was defined as the value of the fit function at its maximum. A trace was classified as 'success' when its amplitude exceeded two standard deviations ($2\sigma$) of baseline noise.

## Long-term imaging of spine morphology

The use of HEPES-buffered sterile-filtered ACSF allowed us to optically stimulate and image slice cultures under near-sterile conditions, using no perfusion system. The custom recording chamber (1 mm quartz glass bottom) and 60 × water immersion objective were sterilized with 70% ethanol and filled with 1.5 ml sterile ACSF. A small patch of membrane (5 × 6 mm) supporting the hippocampal culture was cut out of the cell culture insert (Millipore PICM0RG50), placed in the recording chamber and weighted down with a u-shaped gold wire. During imaging, the temperature of the slice culture was maintained at 25°C via a permanently heated oil-immersion condenser (NA = 1.4, Olympus). After each imaging session, the membrane patch was placed on a fresh sterile membrane insert and returned to the incubator. In the first imaging session, a spine displaying stimulation-induced EPS-CaTs was centered and a three-dimensional image stack (XY: 10 × 10 μm, Z: 5–15 μm) of the mCeru-lean signal was acquired. Additional image stacks were acquired at low magnification to ensure identity of the dendritic segment. For post-hoc analysis of spine turnover, the three-dimensional image stacks were aligned based on a rigid-body algorithm (ImageJ). All spines identified in the three-dimensional image stack acquired before the plasticity induction protocol were analyzed in the subsequent stacks, with the following exception: Spines that appeared shifted from their original position on the dendrite by more than 1 μm in any direction between two consecutive imaging sessions were not included in the analysis, as it was not clear whether the original spine was replaced by a new one. If the imaged neuron showed any sign of compromised health at day 7 (bright GCaMP6 fluorescence at rest, dendritic swelling or beading), the experiment was excluded from the analysis. Maximum intensity projections are shown for illustrative purposes only and were not used for analysis. To estimate spine volume, we integrated the fluorescence intensity of the spine head (mCerulean) taken from a single optical section through the center of the spine. For each spine the point-spread-function (PSF) of the microscope was immersed in the apical trunk of the dendrite to obtain the maximum intensity. In case of different depth of spine and calibration measurement, we corrected for laser attenuation in the tissue. The volume of the PSF was determined with PSFj (*Theer et al., 2014*) using 170 nm fluorescent beads (Invitrogen). Knowing the volume of the PSF and the brightness of a given cell's cytoplasm allowed us to convert spine intensity into absolute spine volume (*Svoboda et al., 1996*).

## Statistics

All statistical analysis was performed using GraphPad Prism 6.0. Data were tested for Gaussian distribution by D'Agostino and Pearson omnibus normality test. Normally distributed data were tested for significant differences with a two-tailed t-test (*Figure 3A*) or one-way repeated-measures analysis of variance (ANOVA) followed by Sidak's multiple comparisons test (*Figure 3B,C*). Data with non-normal distribution data were tested with the following nonparametric tests: Two-tailed Wilcoxon matched-pairs signed rank test (*Figures 2A,D* and *3A*), Friedman test followed by Dunn's multiple

comparison test (*Figure 2C*; *Figure 2—figure supplementary 1A-C*). Investigators were not blinded to the group allocation during the experiments. Data analysis was automated as much as possible to preclude investigator biases. All experiments were done with interleaved controls; pharmacological treatments were mixed with untreated cultures.

## Acknowledgements

We thank Iris Ohmert and Sabine Graf for excellent technical assistance, Christian Schulze for modifications of ScanImage software, and Ingke Braren and the viral vector core facility of the University Medical Center Hamburg-Eppendorf for the production of rAAV. This study was supported by the Deutsche Forschungsgemeinschaft DFG through Research Unit FOR 2419 (P4 and P7), Priority Programs SPP 1665 and SPP 1926, Collaborative Research Center SFB 936 (B7), and the European Research Council (ERC-2016-StG 714762).

## Additional information

### Funding

| Funder | Grant reference number | Author |
|---|---|---|
| Deutsche Forschungsgemeinschaft | FOR 2419 | J Simon Wiegert<br>Christine Elizabeth Gee<br>Thomas G. Oertner |
| Deutsche Forschungsgemeinschaft | SFB 936 / B7 | Thomas G. Oertner |
| Deutsche Forschungsgemeinschaft | SPP 1665 | Thomas G. Oertner |
| European Commission | ERC-2016-StG 714762 | J Simon Wiegert |
| Deutsche Forschungsgemeinschaft | SPP 1926 | J Simon Wiegert |

The funders had no role in study design, data collection and interpretation, or the decision to submit the work for publication.

### Author contributions

J Simon Wiegert, Conceptualization, Data curation, Software, Formal analysis, Supervision, Investigation, Methodology, Writing—original draft, Writing—review and editing; Mauro Pulin, Data curation, Formal analysis, Investigation; Christine Elizabeth Gee, Supervision, Writing—review and editing; Thomas G Oertner, Conceptualization, Supervision, Funding acquisition, Methodology, Writing—original draft, Project administration, Writing—review and editing

### Author ORCIDs

J Simon Wiegert http://orcid.org/0000-0003-0893-9349
Mauro Pulin http://orcid.org/0000-0001-6255-0276
Christine Elizabeth Gee http://orcid.org/0000-0003-0345-3665
Thomas G Oertner http://orcid.org/0000-0002-2312-7528

### Ethics

Animal experimentation: Animal procedures were in accordance with the guidelines of local authorities and Directive 2010/63/EU.

### Decision letter and Author response

Decision letter https://doi.org/10.7554/eLife.39151.019
Author response https://doi.org/10.7554/eLife.39151.020

## Additional files

### Supplementary files
• Transparent reporting form
DOI: https://doi.org/10.7554/eLife.39151.017

### Data availability
All data generated or analysed during this study are included in the manuscript and supporting files. Source data files have been provided for Figures 2, 3, 4 and 5.

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
