## [Decision Letter]

Thank you for submitting your article "The fate of hippocampal synapses depends on the sequence of plasticity-inducing events" for consideration by *eLife*. Your article has been reviewed by three peer reviewers, one of whom is a member of our Board of Reviewing Editors, and the evaluation has been overseen by Gary Westbrook as the Senior Editor. The reviewers have opted to remain anonymous.

The reviewers have discussed the reviews with one another and the Reviewing Editor has drafted this decision to help you prepare a revised submission.

Summary:

This paper presents information that uses a technical advance, and in this way is able to address issues not previously tested. The authors examined the influence of synaptic plasticity-inducing protocols on the long-term stability of dendritic spines of hippocampal CA1 neurons in organotypic slice culture and provide new data regarding the consequences of repeated plasticity-inducing protocols on long-term stability of excitatory synapses in the hippocampus. The question addressed is significant, as the mechanisms that regulate the strength and stability of synapses may be critically important for learning and memory. Several labs have reported previously the LTP-induced stabilization and LTD-induced destabilization of spine synapses, yet only a few have provided limited information regarding the consequences of repeated plasticity induction.

However, the reviewers have several significant concerns and consider that further information is required before the paper is acceptable for publication. The main points for suggested revision are outlined below.

Essential revisions:

1) LTP and LTD experiments need a baseline. A single point of data before induction (or worse, no pre-induction measure at all) is not acceptable. Synaptic strength tends to drift stronger or weaker even without manipulation, due to extraneous factors, and therefore demonstration of a baseline, stable over time, is absolutely required for such plasticity experiments. Without a convincing baseline, one never knows if the strength of the synapses at time of later measurement would have ended up where it did, even without an attempt to induce plasticity.

2) In order to better understand the mechanistic basis of their results, data on the response characteristics and fate of synapses on unstimulated dendritic branches is an essential control for their experiments – the authors have reported this data previously, it should also be included here. Furthermore, raw data is missing from Figure 1 for the time point immediately after oTFS and for the neighbors at all time points (pre, during, immediately post and 30 min post). Summarized data is missing from Figure 2 for both near and distant neighbors. This would be very useful data in interpreting the possible mechanisms that drive the differential stability of nearest versus distant neighbors.

3) The conclusion as to whether it is more difficult to induce LTD 24 hrs after LTP is not well-controlled. Could it also be more difficult to induce LTD on inactive dendrites at 24 hrs? Does the difficulty to induce LTD have anything to do with prior LTP or would prior 0.1 Hz stimulus also interfere with LTD induction? These controls are very important to make that conclusion, and may help the authors provide some mechanistic insight for their findings.

4) A deeper analysis of the survival rate of spines that saw CSBs during induction, but failed to reach the author's criterion for LTP would be an important addition to the paper. They have shown an effect of activity in other experiments. Thus here they should see if they can rule out that the induction stimulus is without effect, even when it does not induce plasticity.

5) It is curious that spines that are depressed, but not below the pre-LTP value, do not change their survival. Are they suggesting that the survival depends not on the plasticity per se, but rather the absolute value of the EPSC, or alternately, that the spine has a memory of what its previous baseline is?

---

## [Author Response]

Essential revisions:1) LTP and LTD experiments need a baseline. A single point of data before induction (or worse, no pre-induction measure at all) is not acceptable. Synaptic strength tends to drift stronger or weaker even without manipulation, due to extraneous factors, and therefore demonstration of a baseline, stable over time, is absolutely required for such plasticity experiments. Without a convincing baseline, one never knows if the strength of the synapses at time of later measurement would have ended up where it did, even without an attempt to induce plasticity.

All our experiments have a baseline. We apologize for not having presented the data in a clearer way. The “single point of data” is in fact the *average* of typically 10-20 stable baseline trial responses acquired over several minutes. To illustrate our evaluation procedure, we added a supplementary figure showing single data points of the example experiment shown in Figure 1. In all experiments, we sampled a stable baseline before we applied oTFS (or oLFS) and recorded EPSCaTs immediately and 30 min after oTFS. (Experiments in which the amplitude of baseline responses fluctuated were immediately terminated.) The averages of each time period were calculated and plotted, resulting in 3 data points per spine, as shown in Figure 2C, for example. This procedure is now exemplified in Figure 1—figure supplement 2.

The plots that apparently lack a baseline are in fact normalized to the baseline value. We originally decided to omit the baseline point as it would always be 1.0 after normalization. We see now that this is visually not intuitive and might be confusing for the reader. In the revised version, we now plot the baseline (average of 10-20 trials, used as normalization value) in Figure 2A, D and E, and Figure 3A. Normalization to the average baseline response is standard practice in LTP experiments.

We added an explanation to the Results section:

*“*To estimate changes in EPSCaT and EPSC amplitude, 10-20 successive traces before and after stimulation were analyzed and averaged (Figure 1—figure supplement 2). EPSCs, integrating the activity of many synapses, showed considerably lower trial-to-trial variability (no failures) compared to EPSCaTs.”

Moreover, to address the important issue as to whether a control synapse would have ended up where an LTP-synapse did, we acquired a new dataset where we recorded EPSCaTs, spine volume and survival after 7 days at synapses that did not receive oTFS. In this condition, synapses did neither change their volume nor their strength over time, and their survival rate was close to unstimulated control spines. This new data is shown in Figure 2—figure supplement 2 and in Figure 3G.

2) In order to better understand the mechanistic basis of their results, data on the response characteristics and fate of synapses on unstimulated dendritic branches is an essential control for their experiments – the authors have reported this data previously, it should also be included here.

We agree that this information would be very valuable. However, due to the increased light pulse intensity during the oTFS protocol (compared to the intensity of ‘test pulses’ to assess synaptic properties), we cannot exclude the possibility that spines on branches that were not imaged during the induction protocol were directly activated during induction. We only have information about the branch harboring the responding spine (see Figure 1B). This was different in our previous paper on the long-term consequences of LTD: LTD induction does not require spikes in the postsynaptic neuron; therefore, we did not step up the stimulation intensity (i.e. light pulse intensity) during the LTD induction protocol.

We added an explanation to the Discussion:

“Since we increased the optogenetic drive to CA3 during oTFS, we could not map the position of all spines that were active during plasticity induction. Therefore, we were not able to study the spatial extent of spine destabilization, e.g. by selecting a ‘control’ branch that received no input during oTFS.”

To address the question how responding spines behave in absence of oTFS, we performed a new set of experiments. We recorded EPSCaTs, spine morphology and long-term survival of spines on dendritic branches (CA1) that were optically stimulated, contained at least one responding spine, but did not receive the oTFS protocol. In this condition synapses did neither change their volume nor their strength over time and their survival rate was close to unstimulated control spines. This new data is shown in Figure 2—figure supplement 2 and in Figure 3G.

Furthermore, raw data is missing from Figure 1 for the time point immediately after oTFS and for the neighbors at all time points (pre, during, immediately post and 30 min post).

We added the missing raw data to Figure 1.

Summarized data is missing from Figure 2 for both near and distant neighbors. This would be very useful data in interpreting the possible mechanisms that drive the differential stability of nearest versus distant neighbors.

For the neighboring spines, we provide information about their volume and 7-day survival. We have no functional information about their strength (LTP/LTD), as we used optogenetic stimulation of ‘connected’ spines. We cannot test the strength of spines that do not receive input from the subset of CA3 cells that expressed channelrhodopsin (By definition, neighbors are spines around a responding spine that did not respond to test pulses). Other studies have used glutamate uncaging to induce plasticity and to probe synaptic strength. In contrast to presynaptic optogenetic stimulation, this method allows to probe any spine at will. However, these experiments are usually performed in zero Mg^2+^ and TTX, which makes optogenetic stimulation impossible (we need action potential propagation). Thus, physiological activation of spine synapses via presynaptic release is a particular strength of our study, but limits functional measurements to optogenetically connected synapses.

Advantages and limitations of our optogenetic approach vs. uncaging stimulation are now explicitly mentioned as follows:

Introduction: “We based our assessment of synaptic strength changes on the amplitude and probability of spine calcium transients (EPSCaTs). […] Compared to glutamate uncaging experiments, which only report changes in postsynaptic strength (potency), optogenetic interrogation is also sensitive to presynaptic changes (release probability), providing a more complete picture of synaptic transmission.”

Results subsection “Optical theta frequency stimulation induced LTP at Schaffer collateral synapses”, last paragraph: “As our functional assessment was limited to the few spines that were synaptically connected to ChR2-expressing CA3 neurons, we could not test whether neighboring enlarged spines were also functionally potentiated.”

3) The conclusion as to whether it is more difficult to induce LTD 24 hrs after LTP is not well-controlled. Could it also be more difficult to induce LTD on inactive dendrites at 24 hrs? Does the difficulty to induce LTD have anything to do with prior LTP or would prior 0.1 Hz stimulus also interfere with LTD induction? These controls are very important to make that conclusion, and may help the authors provide some mechanistic insight for their findings.

As we explain above (Point #2), we cannot identify unstimulated branches after the LTP induction protocol as we used higher light intensities during induction. This we spell out more clearly in the revised discussion. We are quite sure the difficulty to induce LTD is due to the induction of LTP 24 h earlier, and we added the following section to the manuscript to discuss alternative explanations we considered, but eventually ruled out:

Subsection “Effects of sequential plasticity-inducing protocols on synaptic lifetime”: “When oLFS was applied 24 h later, we observed that LTD was induced in only 33% (6/18) of previously potentiated synapses on the next day (Figure 5B), a much lower success rate than the 70% when oLFS was applied with no prior plasticity. […] These results point to a synapse-specific memory of past potentiation events that cannot be detected as increased spine volume, increased release probability or increased EPSCaT potency (see Figure 3).”

The 0.1 Hz chronic stimulation results we decided to remove from the manuscript, as additional controls suggested that in some of these experiments, we may have triggered spikes in postsynaptic neurons, too.4) A deeper analysis of the survival rate of spines that saw CSBs during induction, but failed to reach the author's criterion for LTP would be an important addition to the paper. They have shown an effect of activity in other experiments. Thus here they should see if they can rule out that the induction stimulus is without effect, even when it does not induce plasticity.

The condition the reviewer asks about is untypical and very rare. In the 7-day survival experiments presented in Figure 3, 11 out of 14 oTFS experiments yielded CSBs and *all CSB-spines showed LTP*. Therefore, we have no long-term survival data on spines that *experienced CSBs without reaching LTP*. Of all 41 spines experiencing oTFS (including spines from acute or 1-d experiments which were not followed-up by chronic 7-d imaging), only 6 spines (15% ) showed CSB without subsequent potentiation. And these were the spines with the lowest CSB magnitude (see Figure 2B). Given these odds, getting at least 6 of such spines for a 7-day survival experiment would require 40 oTFS experiments with long-term imaging, which is beyond the scope of this study.

To answer the question whether *the induction stimulus* is without effect, we can consider all spines that received oTFS without experiencing CSBs from the experiment shown in the dusky pink section in Figure 4C. These spines were stimulated, but did not show LTP. If the induction stimulus in itself had a stabilizing effect, these spines should have had increased survival after LTD. This was not the case. In contrast to spines that underwent LTP, they were eliminated with the same probability as spines that had experienced LTD only (Figure 4A). We now spell out this point as follows:

Subsection “Effects of sequential plasticity-inducing protocols on synaptic lifetime”: “We also considered spines that did not experience CSBs to assess whether the oTFS protocol itself would affect synapse lifetime independently of successful LTP induction. […] Without LTP, the oTFS stimulation protocol by itself had no measurable effect on the survival of previously depressed synapses.”

What we cannot rule out from this dataset is that failure to induce LTP might have a destabilizing effect on naive synapses. As we now show in Figure 2—figure supplement 1C, oTFS yielded depression when *not* triggering CSBs. We therefore analyzed survival of such synapses and found that in 50% (3/6) of the experiments spines were eliminated after 7 days – just like after oLFS. Furthermore, the elimination rate of neighbors was also similar to LTD conditions. This suggests that oTFS, which does not lead to LTP, will induce LTD and subsequent spine loss instead. We show the new data in Figure 3F (survival) and included the Ca^2+^ and spine volume data measured on day 0 in Figure 2—figure supplement 1.

This oTFS-LTD did not increase the probability for the elimination of already depressed synapses any further (Figure 4D), suggesting common pathways (occlusion) or a saturation effect.

The new analysis of oTFS-spines that did not get potentiated is mentioned in subsection “Effect of long-term potentiation on synaptic lifetime” as follows:

“As a control, we also analyzed oTFS experiments in which no CSBs were elicited in CA1 neurons. […] This destabilizing effect was contingent on 5 Hz presynaptic activation, as spines that were not stimulated at all or only stimulated by test pulses (responsive spines) had higher survival rates (Figure 3G).”

5) It is curious that spines that are depressed, but not below the pre-LTP value, do not change their survival. Are they suggesting that the survival depends not on the plasticity per se, but rather the absolute value of the EPSC, or alternately, that the spine has a memory of what its previous baseline is?

We assume that this comment refers to the data shown in Figure 5B. To clarify, we recorded a new baseline at day 1, since EPSCaT amplitude can change from day to day (as we show in Figure 3B). In fact, similar to the data shown in Figure 3B, also in this dataset, EPSCaTs did not show consistent potentiation one day after LTP induction. Thus, the relative EPSCaT amplitudes shown for “day1 – LTD” are normalized to a new baseline acquired before the oLFS protocol. However, the spines clearly had a memory of day 0, as most showed potentiation in response to our 1 Hz ‘LTD’ protocol. Naive spines are reliably depressed by this protocol (Figure 4A and C), but not spines that were potentiated one day before. Therefore, the high survival rate (green box in Figure 5B) is most likely explained by our inability to induce LTD at these spines (Figure 5—figure supplement 1C). Essentially, they were twice potentiated, albeit by different stimulation protocols.

Nevertheless, the reviewer makes an excellent point here by suggesting that not the sign of plasticity per se, but the absolute strength of the synapse may be determining spine survival. To test this possibility, we reanalyzed our dataset presented in Figure 5B, asking whether the baseline EPSCaT amplitude on day 0 or day 1 could predict spine survival. Consistent with our previous study (Supplementary Figure 7 in Wiegert and Oertner, 2013), we found that absolute EPSCaT amplitude did not predict survival. We discuss this finding in subsection “Effects of sequential plasticity-inducing protocols on synaptic lifetime” (see below) and show the analysis in Figure 5—figure supplement 1.

“We also considered the possibility that some synapses were already in a depressed state and could therefore not be depressed further. However, the initial EPSCaT amplitude (before oLFS) was not a predictor of successful LTD induction (Figure 5—figure supplement 1C). These results point to a synapse-specific memory of past potentiation events that cannot be detected as increased spine volume, increased release probability or increased EPSCaT potency (see Figure 3).”

“One explanation for the different survival rates could be that the absolute strength of the synapse before the oLFS protocol determined whether it survived, irrespective of the sign of plasticity on day 1 (i.e. the synapse has a memory of its strength and not of its plastic change). However, the strength of the synapse on day 0 or on day 1 did not predict its survival (Figure 5—figure supplement 1D and E), leaving successful induction of depression as the only risk factor we could identify.”